# You Only Train Once: Efficient Tokenizer Selection for Arithmetic in Language Models

**Mucong Ding** [1 2]  **Sean Michael McLeish** [1]  **Kazem Meidani** [2]  **Igor Melnyk** [2]  **Nam H Nguyen** [2]  **C. Bayan Bruss** [2]
**Furong Huang** [1 2]

## Abstract

Tokenization fundamentally shapes how language models perceive and process input, with substantial downstream effects—especially in tasks requiring symbolic or numerical precision. Yet, selecting an optimal tokenizer from a vast design space remains computationally prohibitive, typically requiring full-scale model training for each candidate. Focusing on arithmetic reasoning, we propose **You Only Train Once (YOTO)**, a unified training framework that jointly optimizes the language model and a parameterized distribution over candidate tokenizers. By training a single model using a merged vocabulary and sampling tokenizations adaptively, YOTO enables efficient co-adaptation between model and tokenizer. Applied to arithmetic tasks, YOTO discovers high-performing number tokenizers while dramatically reducing evaluation cost. Our results highlight a promising path toward jointly optimizing tokenizers and models in a principled, scalable manner.

## 1. Introduction

Large Language Models (LLMs) have demonstrated impressive generalization across a range of tasks, yet their performance is tightly coupled to tokenization—the preprocessing step that converts raw input into discrete symbols (Jurafsky and Martin, 2023). As the model's first lens on data, tokenization serves as a perceptual bottleneck, shaping what patterns the model can recognize and learn. Subtle choices here can significantly affect performance, particularly on tasks demanding symbolic precision such as arithmetic, spelling, and multilingual processing (Serrano et al., 2022; Kaddour et al., 2023).

Most modern LMs rely on subword tokenizers like Byte Pair

Encoding (BPE) (Sennrich et al., 2016; Kudo and Richardson, 2018), which are effective for natural language but often brittle in structured domains. For example, numerical inputs are frequently tokenized inconsistently, obscuring magnitude information and hindering arithmetic generalization (Nogueira et al., 2021; Nath et al., 2021).

**The computational challenge of tokenizer selection.** While a wide range of tokenizers have been proposed, there is little consensus on which design choices yield optimal downstream performance. Naively evaluating each tokenizer requires training or significantly finetuning a language model—an expensive process that must be repeated for every candidate. For large-scale models trained from scratch (Guo et al., 2025; Yang et al., 2024a; Grattafiori et al., 2024), this approach becomes computationally infeasible. As a result, tokenizer selection remains intractable at scale. Developing methods that can efficiently surface high-performing tokenizers—without retraining full models per candidate—would mark a significant step forward in scalable model-tokenizer co-design.

**Our contribution: You Only Train Once.** To address the computational bottleneck in tokenizer selection, we propose **You Only Train Once** (YOTO), a unified framework that jointly optimizes a language model and a distribution over candidate tokenizers during pretraining. Rather than training a separate model for each tokenizer, YOTO constructs a merged vocabulary spanning all candidates and trains a *single* model. A learnable distribution governs which tokenizer is applied to each training instance, allowing the model and tokenizer to co-adapt dynamically. This framework enables efficient discovery of high-performing tokenizations in sublinear time.

**Arithmetic reasoning as a testbed.** While our method is broadly applicable, we focus on arithmetic reasoning as a testbed for evaluating tokenizer selection. Arithmetic tasks present a challenging domain where tokenization has a pronounced impact on performance. Standard subword tokenizers frequently fragment numbers inconsistently, obscure magnitude information, and impede gener-

---

[1]University of Maryland, College Park [2]Capital One. Correspondence to: Mucong Ding <mcding@umd.edu>.

*Non-archival presentation at ICML 2025 Tokenization Workshop (TokShop)*, Vancouver, Canada. 2025.

alization (Nogueira et al., 2021; Nath et al., 2021). Prior work has proposed digit-wise (Touvron et al., 2023), block-wise (Yang et al., 2024c;b), and specialized schemes (Golkar et al., 2023), but no consensus exists on a best strategy. Even small differences—such as encoding 1000 as $[100, 0]$ versus $[1, 000]$ (Rando, 2024)—can lead to divergent behavior. This makes arithmetic a well-scoped, high-sensitivity setting for systematically evaluating tokenizer design.

Our contributions can be summarized as:

(1) **Unified Framework for Tokenizer Optimization.** We propose YOTO, a novel training-time objective that jointly optimizes a language model and a parameterized distribution over tokenizers. By sharing a merged vocabulary across candidates, YOTO enables efficient co-adaptation without retraining separate models for each tokenizer.

(2) **Empirical Validation on Arithmetic Reasoning.** We instantiate YOTO on arithmetic tasks, demonstrating that it consistently discovers high-performing number tokenizers at a fraction of the compute cost required by naive approaches. Our results reveal insights into the interplay between tokenizer structure and numerical generalization (see Section 4.2).

## 2. Related Work

### 2.1. Efficient Hyperparameter Optimization for LLM Tokenizers

Optimizing Large Language Models involves tuning numerous hyperparameters, optimizing these is known to be computationally intensive. Research into efficient hyperparameter optimization for deep learning, particularly for language models, can be viewed from several standpoints. Firstly, early termination methods, multi-fidelity optimization techniques like Successive Halving (SHA) (Jamieson and Talwalkar, 2016) and HyperBand (Li et al., 2018) offer principled ways to prune less promising configurations early (Falkner et al., 2018; Wang et al., 2023), assuming reasonable performance correlation across fidelities. Secondly, proxy model transfer, Approaches like $\mu$Transfer (Yang and Mahoney, 2022) aim to predict optimal hyperparameters for large models from smaller ones, though transferring discrete structural choices like tokenizers reliably is challenging (Nguyen et al., 2023; Mahoney et al., 2024). Finally, sequential methods like multi-armed bandits (Audibert et al., 2010), advanced Bayesian Optimization for combinatorial spaces (Baptista and Poloczek, 2018; Oh et al., 2019), and gradient-based methods using differentiable relaxations (Jang et al., 2017; Lorraine et al., 2020; Liu et al., 2018) exist but face scalability or applicability challenges for complex structures. Tokenizer selection can be viewed as a particularly challenging instance of hyperparameter optimization due to its discrete nature, vast combinatorial design space, and

fundamental impact on the model's input data distribution (Feurer and Hutter, 2019). The limitations across these general optimization strategies highlight the need for methods tailored specifically to the efficient optimization of LLM tokenizers.

### 2.2. Number Tokenization Strategies in Prior Work

Representing numerical data effectively is crucial for LLM quantitative reasoning (Nath et al., 2021). Various strategies have emerged to address the shortcomings of standard tokenizers which broadly fit into two categories: changing the segmentation of numbers in tokenization or changing the formatting or positional information passed to the model.

**Segmentation and Chunking.** This concerns how number strings are divided into tokens. Key approaches include: BPE, N-digit chunking or highly specialized schemes. Standard BPE leads to inconsistent segmentation of numbers and hinders learning (Nogueira et al., 2021; Yang et al., 2024c). Employing *N-digit chunking* (e.g., 1-, 2-, or 3-digits) which trades off sequence length and vocabulary size (Yang et al., 2024c;b). Golkar et al. (2023) propose *xVal*, which treats numbers holistically via a single numerical token and decodes the value for this token using a separate head.

**Formatting and Positional Representation.** These techniques modify the input string or embeddings to aid interpretation. Notable strategies involve: reversal, padding, and additional positional encodings. Addition and multiplication begin with the least significant digit; this conflicts with the causal masking in decoder transformers, and reversing the input numbers significantly aids arithmetic performance (Singh et al., 2024; Lee et al., 2023). Moreover, aligning digits of the same significance can be difficult task for transformers. This can be addressed from two angles, firstly, we can zero-pad to fixed length (Shen et al., 2023) or we can pass additional information to the model to descibe this information. For example, Index Hints (Zhou et al., 2023) or Abacus Embeddings (McLeish et al., 2024; Cho et al., 2024a;b), which explicitly encode place value or position.

This array of techniques underscores the complexity of number tokenization. While specific methods show promise, the interactions between choices are intricate, and efficiently finding the optimal combination remains an open challenge (Yang et al., 2024b).

## 3. Experimental Setup

### 3.1. Tokenization for Arithmetic

To ground our investigation into tokenization for arithmetic reasoning, we specify a structured design space for number tokenization, focusing on arithmetic tasks. This space is

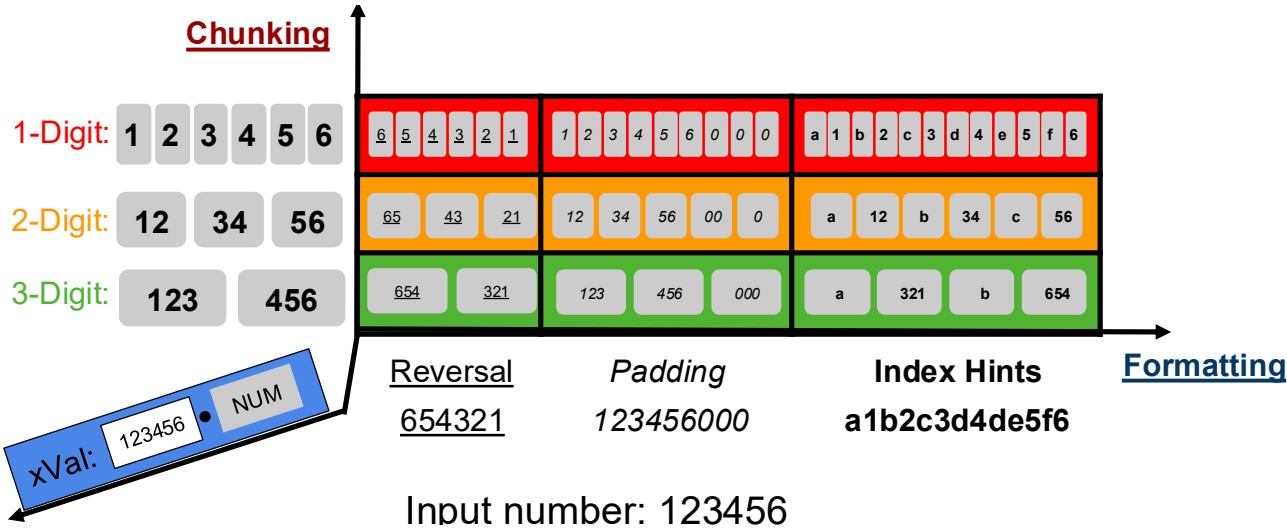

*Figure 1.* **Number Tokenization Strategies Explained.** The 13 strategies analyzed comprise combinations of 3 chunking methods (1, 2, 3-Digit), 3 formatting techniques (Reversal, Padding, Index Hints), and the *xVal* approach. Their encoding of '123456' is illustrated.

constructed by combining choices along three primary axes, based on common strategies discussed previously:

(1) **Representation:** How the number itself is fundamentally represented. We consider two options: standard positional *Integer* and *Scientific Notation*.

(2) **Chunking:** How the digit string (or mantissa/exponent) is segmented. We include four options: fixed *1-digit*, *2-digit*, or *3-digit* chunks, and the specialized *xVal* approach (Golkar et al., 2023) which treats the whole number as a single token.

(3) **Ordering/Formatting/Indexing:** Additional techniques applied to the segmented representation. We consider four mutually exclusive options: *None*, most significant digit first; *Reverse*, least significant digit; *Padding*, padding each number with zeros to a fixed length, or *Index Hints*, specialized tokens/embeddings to represent the significance of a digit. We assume at most one of these techniques is active for any given tokenizer configuration.

We focus on integer addition and scientific notation of multiplication, considering the 3 chunking strategies and 4 formatting strategies we gain 12 experiments as we also run each chunking strategy without any formatting additions. We also analyze *xVal* totaling 13 distinct experiments to analyze. This structured space (see Figure 1) allows for systematic evaluation and analysis of the interactions between different design decisions.

We focus our experiments on addition and multiplication. For each setting, we generate synthetic datasets consisting of problems formatted as $operand\_1 \; operator \; operand\_2 = result$ (e.g., $A + B = C$ or $A \times B = C$), where operands $A$ and $B$ are presented in the integers or in scientific nota-

tion. We follow Yang et al. (2024b) to generate datasets to ensure controlled operand ranges and result distributions suitable for evaluating numerical understanding. Balancing computational feasibility and statistical significance, we use datasets of $100k$ training samples and $10k$ test samples.

### 3.2. Models and Baselines

We employ Transformer-based architectures scaled systematically based on the Super Tiny LMs principles (Hillier et al., 2024). We mainly use the $104M$ sized model. While studying the scaling effects as ablation studies, we may use LMs with sizes ranged from approximately $10M$ up to $104M$ parameters (e.g., $10M, 18M, 26M, 38M, 50M, 104M$). Models are trained from scratch for each experiment unless otherwise noted, such as within the joint training framework which uses a single shared model.

We evaluate model performance on the arithmetic tasks using three primary metrics: exact match, mean absolute error, and mean relative error. Exact match accuracy is the percentage of problems where the models output exactly matches the ground truth. Mean absolute error is the average numerical difference between models output and ground truth. Mean relative error is the mean absolute error divided by the absolute value of the ground truth.

The primary baseline for comparison is the ground truth evaluation. This involves independently training a dedicated model from scratch for each candidate tokenizer. This baseline represents the standard, computationally expensive approach to tokenizer selection, establishing the target performance ranking and a computational cost. We show

our efficient optimization method surpasses this baseline by measuring the computational efficiency (number of Floating Point Operations) required to find the best tokenization and Spearman's Rank Correlation Coefficient between the tokenizer selected by YOTO and the baseline evaluations.

### 3.3. Method

YOTO introduces a novel framework for efficiently determining optimal tokenizer configurations by jointly optimizing tokenizer choices and language model parameters. This approach, visualized in Figure 2, is built upon three core principles:

(1) **Unified Vocabulary and Shared LLM:** We train a single Large Language Model, parameterized by $\theta$, whose vocabulary is a comprehensive union of all tokens from all candidate tokenizers. This significantly amortizes the computational cost typically associated with training separate models for each tokenizer configuration.

(2) **Parameterized Exploration of Combinatorial Tokenizer Space:** The selection of a tokenizer is governed by a learnable categorical distribution, parameterized by $\varphi$. This parameterization allows for efficient exploration of the often vast and combinatorially structured design space of tokenizers, guiding the search towards optimal configurations.

(3) **Resource-Aware Early Termination via Soft Successive Halving:** To further enhance efficiency, YOTO integrates a soft variant of the Successive Halving Algorithm. This mechanism dynamically allocates training resources, prioritizing promising tokenizer candidates and prematurely terminating unpromising ones, thereby reducing the overall computational budget.

The complete YOTO algorithm is summarized in Algorithm 1.

**Joint optimization of model and tokenizer distribution parameters underpins our methodology.** As illustrated in Figure 2, the training process jointly optimizes LLM parameters $\theta$ and tokenizer sampling distribution parameters $\varphi$. Input data is processed in mini-batches. For each instance $x$, a tokenizer $S$ is sampled from $p(S|\varphi)$ over the candidate set $\mathcal{T}$. Instance $x$ is then tokenized by $S$ into a token ID sequence, which is fed to the LLM. The LLM's embedding layer, designed for the unified vocabulary, maps tokens from any $S$ to corresponding embeddings. The LLM then performs its primary task (e.g., next-token prediction), and a loss $\mathcal{L}_{\text{LLM}}(x, S, \theta)$ is computed.

**The overall loss guides updates to both LLM and tokenizer distribution parameters.** The mini-batch loss $\hat{\mathcal{L}}(\theta, \varphi)$ empirically approximates the expected loss over

data and tokenizer distributions:

$$\hat{\mathcal{L}}(\theta, \varphi) = \frac{1}{|B|} \sum_{x_i \in B} \mathcal{L}_{\text{LLM}}(x_i, S_i, \theta), \quad \text{where } S_i \sim p(S|\varphi).$$

This loss depends on both $\theta$ and $\varphi$. During backpropagation, gradients update both parameter sets. LLM gradients $\nabla_\theta \hat{\mathcal{L}}$ are standard. To obtain gradients $\nabla_\varphi \hat{\mathcal{L}}$ for the discrete tokenizer sampling, we use the Gumbel-Softmax reparameterization (Jang et al., 2017; Maddison et al., 2017). This technique enables differentiable sampling from $p(S|\varphi)$ by introducing Gumbel noise and a temperature $\tau$, allowing gradient backpropagation to update $\varphi$. Specifically, if $\varphi$ parameterizes logits $\alpha_S$ per tokenizer $S \in \mathcal{T}$, Gumbel-Softmax provides a continuous, differentiable approximation to discrete sampling, facilitating gradient-based optimization of $\varphi$. For smaller tokenizer design spaces, Bayesian optimization can be an effective gradient-free alternative for optimizing $\varphi$.

**Efficient resource management is achieved via a soft adaptation of Successive Halving.** To enhance computational efficiency and expedite optimal tokenizer discovery, YOTO incorporates a soft adaptation of the Successive Halving Algorithm (SHA) (Jamieson and Talwalkar, 2016). SHA iteratively allocates resources, progressively pruning underperforming configurations—in our case, individual tokenizers within the support of $p(S|\varphi)$. Key SHA hyperparameters include the initial budget $r_0$ per candidate, a reduction factor $\eta > 1$, and the maximum budget $R_{max}$ per tokenizer. SHA proceeds in rungs: in rung $k$, active candidates $\mathcal{T}_k$ are evaluated with an additional budget. Based on performance (e.g., validation loss contribution), only the top $1/\eta$ fraction advances to rung $k+1$, receiving $\eta$ times more budget.

**Our soft SHA implementation dynamically manages tokenizer participation through learned logits.** Instead of a hard set of active tokenizers, participation is managed via $\varphi$. If SHA deems a tokenizer $S$ pruned (i.e., budget exhausted and not in the top fraction), its corresponding logit $\alpha_S$ in $\varphi$ is set to $-\infty$, making its sampling probability $p(S|\varphi)$ effectively zero. The distribution $p(S|\varphi)$ is then renormalized over active candidates. Budgets are tracked by monitoring processed data samples per tokenizer. This early termination of less promising designs significantly cuts cumulative training costs by focusing resources on a diminishing set of candidates, which is especially effective as performance differences often emerge early.

**Alternative optimization strategies and vocabulary reuse enhance framework flexibility.** While gradient-based Gumbel-Softmax is the primary described mechanism for optimizing $\varphi$, iterative methods like Bayesian optimization,

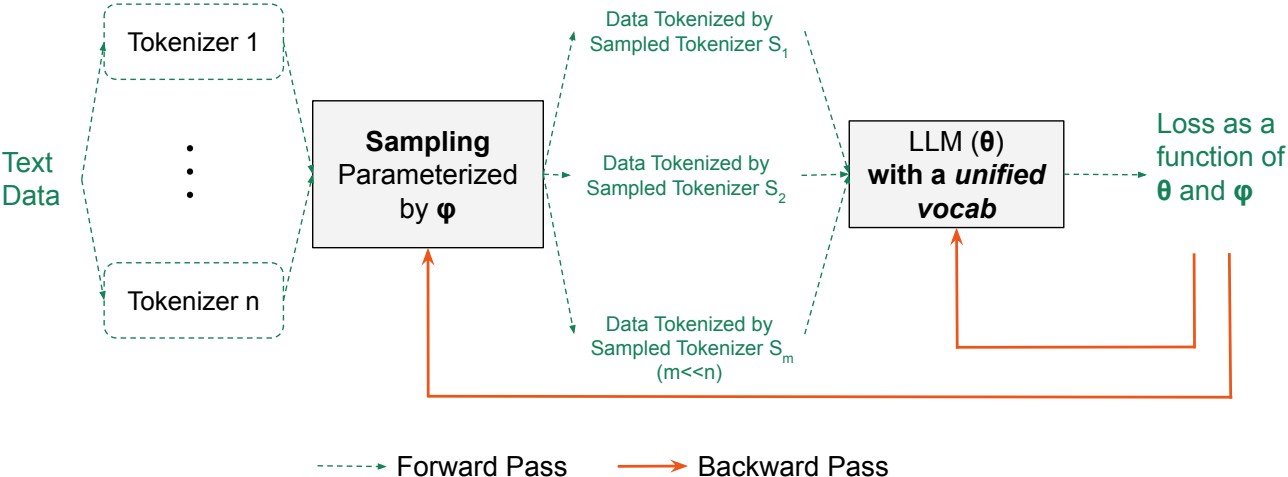

*Figure 2.* **Visualization of You Only Train Once.** We jointly optimize the language model parameters and the parameters we use to sample the tokenizers we are considering. During the forward pass we sample a tokenizer to use for each row of the batch according to the tokenizer sampling distribution, we then pass these through the language model which has the joint vocabulary for all tokenizers. On the backward pass we propagate the loss through both the language model and the tokenizer sampling distribution, updating both at every step.

---

**Algorithm 1** YOTO: You Only Train Once
---
1: **Input:** Corpus $\mathcal{D}$; initial LLM params $\theta_0$; initial tokenizer logits $\varphi_0$ for candidates $\mathcal{T}$; SHA params $(r_0, \eta, R_{\max})$; Learning rates $\lambda_\theta, \lambda_\varphi$; Gumbel temp $\tau$.
2: Initialize: Unified vocab from $\mathcal{T}$; LLM $\theta \leftarrow \theta_0$; tokenizer logits $\varphi \leftarrow \varphi_0$.
3: Initialize SHA: $\mathcal{T}_{\text{active}} \leftarrow \mathcal{T}$; budgets $b_S \leftarrow 0$, performance $m_S \leftarrow 0 \ \forall S \in \mathcal{T}$; rung budget $r_{\text{rung}} \leftarrow r_0$.
4: **for** training step $t = 1, \ldots, T_{\max}$ **do**
5:      Sample batch $B \subset \mathcal{D}$; Let $B_{\text{proc}} \leftarrow \emptyset$.          ▷ *Collects (tokenized sample, $S_x$)*
6:      **for** each sample $x \in B$ **do**
7:          Sample tokenizer $S_x \sim p(S|\varphi, \mathcal{T}_{\text{active}})$.          ▷ *Sample from active distribution*
8:          Add $(\text{Tokenize}(x, S_x), S_x)$ to $B_{\text{proc}}$; $b_{S_x} \mathrel{+}= 1$.      ▷ *Tokenize, and track $S_x$ & its budget*
9:      $\hat{\mathcal{L}} \leftarrow \frac{1}{|B|} \sum_{(x_{\text{tok}}, S_x) \in B_{\text{proc}}} \mathcal{L}_{\text{LLM}}(x_{\text{tok}}, S_x, \theta)$.      ▷ *Batch loss over sampled tokenizers*
10:      Compute gradients $\nabla_\theta \hat{\mathcal{L}}$ and $\nabla_\varphi \hat{\mathcal{L}}$.      ▷ *Compute $\nabla_\varphi$ via Gumbel-Softmax ($\tau$)*
11:      $\theta \leftarrow \theta - \lambda_\theta \nabla_\theta \hat{\mathcal{L}}$; $\varphi \leftarrow \varphi - \lambda_\varphi \nabla_\varphi \hat{\mathcal{L}}$.      ▷ *Update LLM & active tokenizer logits*
12:      Periodically update performance $m_S$ for $S \in \mathcal{T}_{\text{active}}$.      ▷ *Using validation data*
13:      **if** SHA rung evaluation triggered **then**      ▷ *Budget milestones met*
14:          $\mathcal{T}_{\text{eval}} \leftarrow \{S \in \mathcal{T}_{\text{active}} \mid b_S \geq r_{\text{rung}}\}$.      ▷ *Candidates completing current rung*
15:          **if** $|\mathcal{T}_{\text{eval}}| \geq \eta$ **then**      ▷ *Sufficient candidates for SHA step*
16:              Sort $\mathcal{T}_{\text{eval}}$ by $m_S$; $\mathcal{T}_{\text{promoted}} \leftarrow$ top $\lceil |\mathcal{T}_{\text{eval}}|/\eta \rceil$ from $\mathcal{T}_{\text{eval}}$.
17:              **for** $S \in \mathcal{T}_{\text{eval}} \setminus \mathcal{T}_{\text{promoted}}$ **do**
18:                  Set logit $\alpha_S \leftarrow -\infty$ in $\varphi$; $\mathcal{T}_{\text{active}} \leftarrow \mathcal{T}_{\text{active}} \setminus \{S\}$.      ▷ *Mask logits & prune*
19:              $r_{\text{rung}} \leftarrow r_{\text{rung}} \cdot \eta$.      ▷ *Increase budget for next rung*
20:              **if** $r_{\text{rung}} > R_{\max}$ or $|\mathcal{T}_{\text{active}}| \leq 1$ **then**
21:                  **break**.      ▷ *Max budget or few candidates*
22: **Output:** Optimized $\theta^*$, final $\varphi^*$ (or best performing $S^*$).

---

informed by performance $m_S$ from SHA, offer a viable alternative for manageable design spaces, potentially avoiding direct gradient computation for $\varphi$ and management of the Gumbel-Softmax temperature $\tau$. We also emphasize that although YOTO uses a unified vocabulary larger than any single tokenizer's, many underlying token elements (common digits chunks, subwords, characters) are naturally reused across different tokenization strategies. This structure allows for the principled discovery of the best-performing tokenizer from diverse candidates within a single, integrated training paradigm.

# 4. Exploring Tokenization for Arithmetic

We now move to empirically exploring which tokenization, formatting and indexing techniques are most effective for arithmetic and empirically verifying the usefulness of YOTO. In Section 4.1, we explore prior tokenization techniques and tricks to ground our exploration of tokenization methods for arithmetic. In Section 4.2, we show how YOTO discovers the most efficient tokenization techniques for arithmetic using our joint optimization objective. Implementation details and hyperparameter setups are discussed in the Appendix.

## 4.1. Analysis of Ground Truth Tokenization Results

Prior work has extensively explored tokenization and formatting techniques for arithmetic; however, seldom do studies compare many of them in a controlled and equivalent setting.

In Figure 3, we show the test results when training each model from scratch with different tokenization and formatting techniques for addition. As *xVal* only assigns a single token to all numbers, we cannot apply any formatting techniques, hence the formatting columns are empty. On the left of Figure 3, we present exact match accuracy, seeing that smaller digit group tokenization is preferable, and *xVal* performs as well as 2-digit tokenization strategies but struggles to perform as well as single-digit tokenization techniques. We see that reversing numbers and padding leads to a performance increase for all tokenization schemes. For single-digit tokenization, applying these techniques leads to better performance than *xVal*. On the right of Figure 3, we present the mean absolute error and see the same trend as for exact match accuracy.

In Figure 4, we show the results from training each tokenization and formatting strategy on multiplication in scientific notation. We see the same general trend for the digit group tokenization strategies as in Figure 3 for addition, with single digit tokenization performing best. However, for multiplication we see *xVal* performs significantly better than any other technique with a much lower residual error of only 0.1. Numerous studies have identified multiplication as a particularly challenging task (Dziri et al., 2023; McLeish et al., 2024), suggesting that techniques like YOTO may offer greater utility in this context as the community continues to explore the design space.

## 4.2. Joint Training: Effective and Efficient

Now we have a comprehensive set of baseline methods, and we compare our joint training strategy to them. In

Section 4.2.1, we see our joint training method is capable of recovering the trend as to which tokenizer is most efficient to a good degree of accuracy. In Section 4.2.2, we emphasize the gain from our method as it only requires training once to find the optimal tokenizer, hence the name: You Only Train Once!

### 4.2.1. EFFECTIVENESS

In Figure 5 we plot the ranking of the tokenization methods by training a unique model from scratch (x axis) and the ranking found by the YOTO training objective. For convenience, we plot the perfect trend line ($y = x$) in red and annotate Spearman's rank coefficient $\rho$ on each plot.

In Table 1, we show the Pearson correlation coefficient, Spearman's rank correlation coefficient, and Kendall's tau for each of the three settings we study. We see high positive values for all of these coefficients, suggesting a good positive correlation between the best tokenization method found by YOTO and the ground truth. Most importantly, we can visually see in Figure 5 that the best tokenization method is always within the top two found by YOTO. This means that the worst case cost of YOTO is the amount to train YOTO plus 2 runs from scratch, much lower than the naive method of trying all combinations. We emphasize that it is more important for YOTO to have higher fidelity at top rankings than lower rankings, as we want to find optimal tokenization strategies, which we visually see is true in Figure 5.

We see in Figure 5 and Table 1 that YOTO is better at predicting correlations for addition than for multiplication. We suggest this is due to more noise within the system than for addition, as multiplication has been found to be a much more difficult problem to solve than addition (McLeish et al., 2024), and it has even been suggested that generalization in multiplication may be beyond the limits of transformers (Dziri et al., 2023).

### 4.2.2. EFFICIENCY

We train all models on *Nvidia L40s GPUs*, capable of 362.05 TFLOPs in `bfloat16`. However, due to the shared tokenization and early stopping used by the joint training strategy, we achieve significant computational savings.

In Table 2, we show the number of Floating Point Operations (FLOPs) required to train a single baseline model and the number of FLOPs required to train our method.

Firstly, we see that this saving for one model run is very large ($> 80\%$). When we take into account that to thoroughly ablate tokenizer choice one must train a sweep of models from scratch (13 in our case) this saving is many

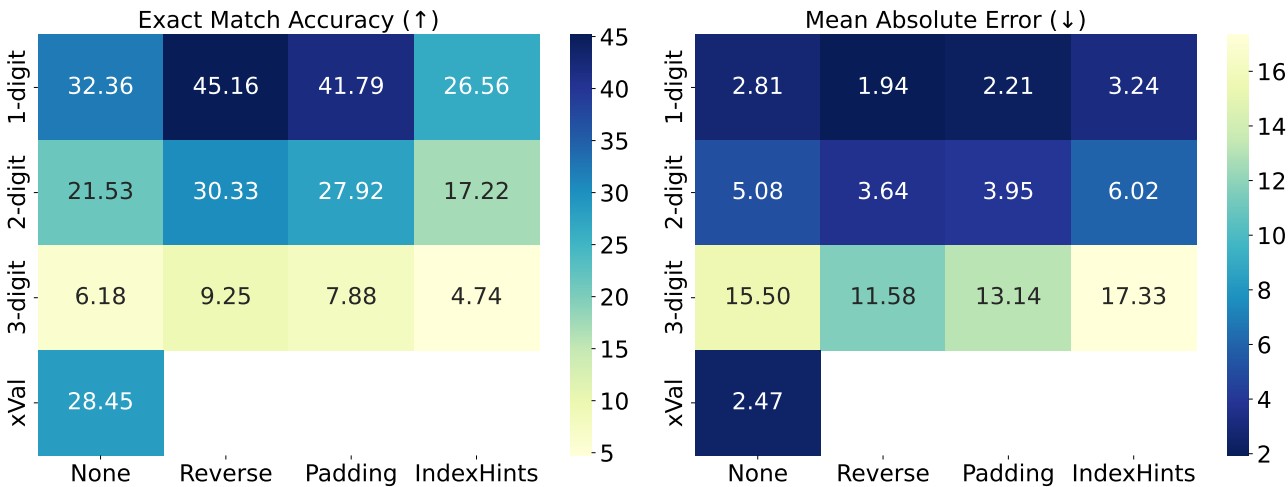

*Figure 3.* **Exact Match Accuracy (left) and Mean Absolute Error (right) for integer addition.** On the y axis we vary the tokenization method used for numbers, on the x axis we vary any additional formatting methods used to aid addition. We see that single digit tokenization performs best when chunking numbers during tokenization and reversing numbers leads to a large gain in performance. Furthermore, we find *xVal* is able to perform approximately as well as a single digit tokenization scheme.

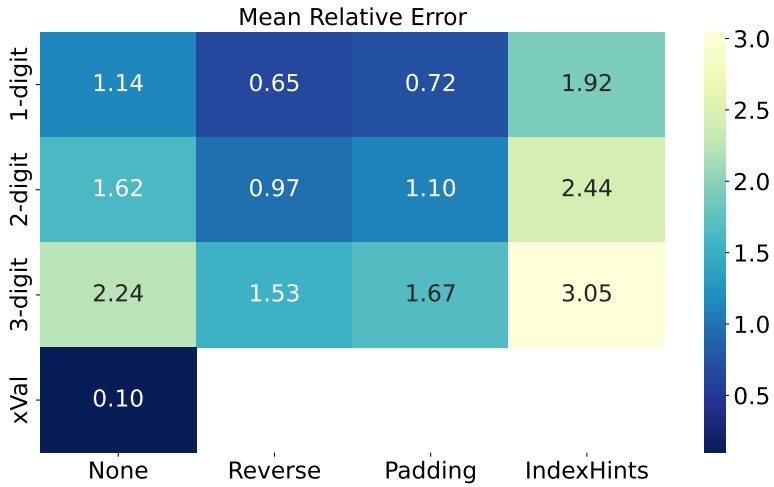

*Figure 4.* **Mean Relative Error for scientific notation multiplication.** On the y axis, we vary the tokenization method used for numbers, on the x axis, we vary any additional formatting methods used to aid multiplication. For the chunking tokenization strategies, smaller chunks offer better performance; however, unlike for addition, we see *xVal* achieves by far the lowest error of all techniques. Relative instead of absolute errors are reported for scientific notation multiplication tasks because of the long-tailed absolute error distribution.

*Table 1.* Correlation metrics between Ground Truth and Joint Training across all data points. We see a high correlation between YOTO and the baseline runs, which are completed by training individual models from scratch.

|  | **Pearson** | **Spearman** | **Kendall** |
|---|---|---|---|
| Addition EMA | 0.899 | 0.912 | 0.769 |
| Addition MAE | 0.950 | 0.918 | 0.744 |
| Multiplication MRE | 0.709 | 0.714 | 0.538 |

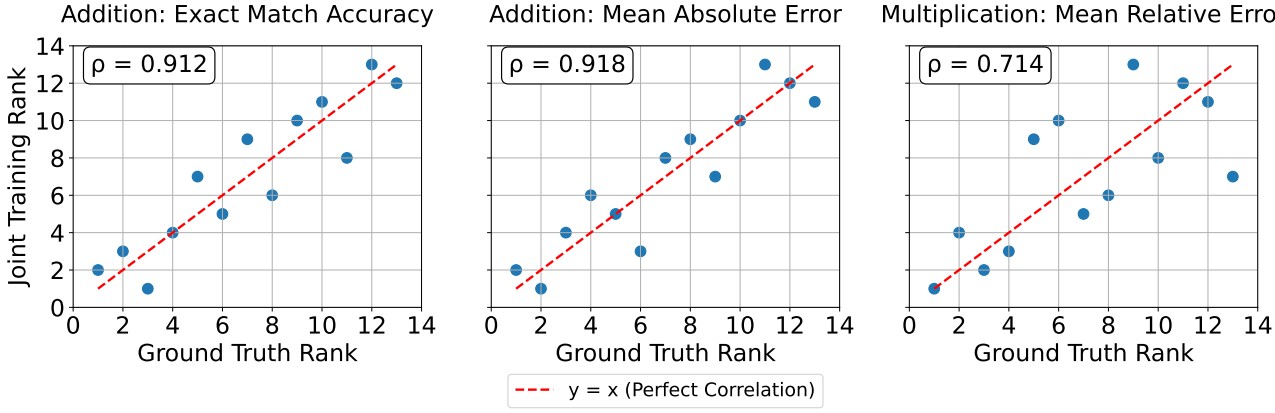

*Figure 5.* **Correlation between baselines and YOTO.** On the y axis, we have the ranking of each tokenization and formatting method according to YOTO, on the x axis, we plot the ground truth ranking, which we record in Section 4.1. We annotate the Spearman's rank correlation coefficient from Table 1 onto each plot, seeing high correlation. Most importantly, we see that the best performing ground truth method is always within the top two found by YOTO.

*Table 2.* We record the number of FLOPs required to train a *single* baseline model compared to the number of FLOPs required to train YOTO. We see the savings are significant and are amplified as practitioners are required to train many more baseline models if not using YOTO.

|  | **Baseline Training (ExaFLOPs)** | **Joint Training (ExaFLOPs)** | **ExaFLOPs saved (%)** |
|---|---|---|---|
| Addition | 22.96 | 3.67 | 84.0 |
| Multiplication | 29.09 | 4.17 | 85.6 |

magnitudes larger in real terms. For example, to find the optimal tokenizer in all cases for the tasks shown in this paper, we would have to train at most two models and the joint training objective model when using YOTO compared to having to train 13 different models to brute force search the space. Moreover, we want to highlight that this time complexity saving increases as the tokenizer search space becomes larger, because while the number of LLM trainings to sweep all tokenizers increases, our YOTO only requires one-time training.

## 5. Conclusion and Future Work

This paper addresses the underexplored but critical problem of tokenizer selection in large language models (LLMs), where vast design spaces and expensive evaluations hinder practical optimization. We introduced YOTO, an efficient framework that jointly learns language model parameters and a distribution over candidate tokenizers. By training a single model over a unified vocabulary and adaptively sampling from tokenization strategies, YOTO eliminates the need to train one model per tokenizer—dramatically reducing computational cost while preserving downstream performance. Applied to arithmetic reasoning, YOTO identifies high-performing number tokenizers and yields actionable

insights into how tokenization impacts generalization.

Our findings suggest several promising directions for future work:

**Scaling to broader tokenizer design spaces.** While this work focused on structured numeric tokenization, our method is general. A natural next step is to extend YOTO to explore larger, more diverse tokenizer search spaces—including morphological, multilingual, or byte-level schemes—where exhaustive comparisons (e.g., (Yang et al., 2024b)) become prohibitively expensive. YOTO provides a scalable engine for accelerating such comparisons.

**Application to larger models and varied modalities.** Scaling YOTO to frontier LLMs (e.g., 13B, 70B+) and applying it beyond arithmetic to general natural language, code, categorical inputs, or even time-series data could unlock further benefits. Different domains may exhibit distinct tokenization sensitivities that co-adaptive training can reveal.

Together, these directions highlight YOTO as a foundational step toward more adaptive and computationally efficient co-design of tokenizers and language models.

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

# Appendices

## A. Limitations

YOTO offers a novel framework for efficient tokenizer selection, showing significant computational savings and strong performance in identifying optimal arithmetic tokenizers. While promising for model-tokenizer co-design, its current scope has limitations that suggest avenues for future work.

**Focused Application to Arithmetic Tokenization.** Our empirical validation centers on arithmetic tasks and number representations. This focus, while potentially seen as niche, was deliberate. It stems not from avoiding general language complexities but from the current state of tokenizer design research. The arithmetic domain provides a uniquely rich, structured landscape of tokenization strategies (e.g., digit-wise, xVal, various formatting), essential for rigorously testing YOTO within a well-defined search space. Extending YOTO to general language tokenization requires developing similar structured design spaces for broader vocabulary elements, a substantial research challenge.

**Model Scale and Training from Scratch.** Our experiments use models up to 100M parameters, smaller than many state-of-the-art LLMs. This scale was chosen because fair tokenizer evaluation necessitates from-scratch training or co-adaptation, as fine-tuning larger models introduces biases. Furthermore, models of this size are sufficient for the arithmetic tasks and datasets used, effectively demonstrating YOTO's algorithmic efficiency in tokenizer selection. While scaling YOTO to larger models is future work, the current scale balances rigorous evaluation with computational feasibility for from-scratch training.

**Scope of Explored Optimization Techniques.** YOTO uses a soft Successive Halving (SHA) variant and Gumbel-Softmax for tokenizer sampling. While effective, exploring broader hyperparameter optimization algorithms for the tokenizer distribution and advanced early-termination methods could yield improvements. The interplay between merged vocabulary size, candidate number, and the optimization strategy for $\varphi$ particularly warrants deeper investigation for larger candidate sets.

## B. Broader Impacts

The YOTO framework, primarily an academic contribution improving a fundamental aspect of LLM pipelines, offers several positive broader impacts. By making tokenizer selection significantly more resource-efficient, it can enhance **development efficiency and accessibility**; potentially lowering entry barriers for creating custom LLMs, especially for those with limited compute, and fostering innovation. This efficiency also facilitates **advancing model performance on specialized tasks**; better tokenizers, crucial for areas like arithmetic, code, or scientific analysis, can lead to more accurate and reliable LLMs in these fields. Furthermore, YOTO may **stimulate research into model-input co-design**; highlighting the benefits of co-adapting tokenizers and models and encouraging more integrated approaches. While YOTO itself is an optimization algorithm, any resulting improved models should be developed responsibly, considering ethical implications common to powerful AI.

## C. Formal Definition of Efficient Tokenization Optimization

Let $\mathcal{T} = \{T_1, \ldots, T_C\}$ be a finite set of $C$ candidate tokenizer configurations derived from a design space. Given a fixed model architecture $M$, a set of training hyperparameters $H$, a training dataset $D_{\text{train}}$, and a validation dataset $D_{\text{val}}$, the goal is to identify the optimal tokenizer $T^* \in \mathcal{T}$. Training the model $M$ with a specific tokenizer $T_i \in \mathcal{T}$ under hyperparameters $H$ on $D_{\text{train}}$ yields optimal parameters $\theta^*(T_i)$. Due to stochastic elements in training (e.g., initialization, data shuffling), performance varies. Let $\mathcal{P}(M_\theta, D)$ denote a performance metric (e.g., accuracy) on dataset $D$ with model parameters $\theta$. We seek the tokenizer $T^*$ that maximizes the expected validation performance:

$$T^* = \arg\max_{T \in \mathcal{T}} \mathbb{E}\left[\mathcal{P}(M_{\theta^*(T)}, D_{\text{val}})\right] \tag{1}$$

where the expectation $\mathbb{E}[\cdot]$ is taken over the sources of training randomness. The naive approach involves independently training the model for each $T \in \mathcal{T}$ and selecting the best based on validation performance. Let $\mathbb{E}[Time_{\text{train}}(M, D_{\text{train}}, H)]$ be the expected wall-clock time for a single training run. The naive evaluation cost is $Time_{\text{naive}} = C \times \mathbb{E}[Time_{\text{train}}]$. The core challenge is to develop an optimization algorithm $\mathcal{A}$ to find $T^*$ (or a top-$k$ set) such that its total runtime, $Time(\mathcal{A})$,

satisfies the efficiency constraint:

$$Time(\mathcal{A}) \ll Time_{\text{naive}} \tag{2}$$

Ideally, $Time(\mathcal{A})$ should scale sub-linearly with $C$.

## D. Rationale for Tokenizer Design Space

The design of an effective tokenizer, particularly for tasks requiring numerical precision like arithmetic, involves navigating a vast space of potential strategies. This section details the rationale and methodology for arriving at the 13 distinct number tokenization configurations evaluated in this work (see main paper, Section 3.1), focusing on the mutual compatibility and interplay of various considered techniques. Our goal was to establish a structured and diverse design space that allows for systematic analysis of model-tokenizer co-adaptation.

We began by identifying several promising low-level design choices for number tokenization, drawing from existing literature and common practices. These initial candidate techniques can be broadly categorized as:

- **Segmentation/Chunking Strategies**:
  - *N-Digit Chunking*: Segmenting number strings into fixed-size chunks of 1, 2, or 3 digits. This is a common baseline approach for number representation (Yang et al., 2024c).
  - *xVal* (Golkar et al., 2023): Treating the entire number as a single, holistic token, with its numerical value decoded by a specialized mechanism.

- **Ordering, Formatting, and Indexing Strategies**: These techniques modify the number string or its representation to aid model interpretation.
  - *Reverse Digits*: Reversing the order of digits in the number string (e.g., "123" becomes "321"). This aligns processing with the least significant digit first, which can be beneficial for arithmetic operations (Singh et al., 2024; Lee et al., 2023).
  - *Zero Padding*: Padding numbers with leading zeros to a fixed maximum length before tokenization (Shen et al., 2023). This can help models align digits of similar significance across different numbers.
  - *Length Prefix (Considered but not in final 13)*: Prepending a special token or sequence indicating the length of the number. This is an alternative to Zero Padding for explicitly providing length information.
  - *Index Hints* (Zhou et al., 2023): Employing specialized tokens or embeddings to explicitly indicate the significance or place value of each digit or chunk.
  - *Abacus Embeddings (Considered but not in final 13)* (McLeish et al., 2024): Using learned embeddings that represent each digit's place value, which are then typically summed to form the number's representation.

These techniques are generally applicable to both integer and scientific notation representations. For scientific notation (e.g., $M \times 10^E$), string-based operations like Reverse Digits or Zero Padding are typically applied to the mantissa $M$, while the exponent $E$ might be handled separately or as part of the overall strategy (e.g., also reversed or padded). Index Hints can be adapted to signify digit positions within both the mantissa and the exponent.

A critical step in defining a practical design space is to assess the mutual compatibility of these techniques. Not all combinations are sensible or offer unique advantages. Table 3 summarizes the pairwise compatibility of these fine-grained design choices.

From this compatibility analysis (detailed in Table 3), we made several key decisions to arrive at the final 13 configurations explored in the main paper:

1. **xVal as a Standalone Strategy**: As *xVal* tokenizes the entire number into a single conceptual unit, it is fundamentally incompatible with techniques that operate on individual digits or sub-digit chunks (e.g., Reverse Digits, N-Digit Chunking, Index Hints). Thus, xVal forms one distinct branch in our design space, used without other formatting or indexing augmentations.

2. **N-Digit Chunking as a Base**: For more granular tokenization, we selected *N-Digit Chunking* (with N=1, 2, or 3) as the foundational segmentation approach. The remaining formatting and indexing techniques are considered as potential augmentations to these chunking strategies.

*Table 3.* Mutual Compatibility of Number Tokenization Design Choices. (✓ denotes Compatible; × denotes Incompatible or highly redundant/conflicting; △ denotes Conditionally compatible, may require non-trivial adaptation, or offers overlapping functionality that might be better addressed by one technique alone.) N/A indicates self-comparison or inherent redundancy within a conceptual pairing.

| Technique/Type | N-Digit Chunking | xVal | Reverse Digits | Zero Padding | Length Prefix | Index Hints | Abacus Emb. |
|---|---|---|---|---|---|---|---|
| **N-Digit Chunking** | N/A | N/A | ✓ | ✓ | ✓ | ✓ | ✓ |
| **xVal** | N/A | N/A | × | × | × | × | × |
| **Reverse Digits** | ✓ | × | N/A | ✓ | ✓ | ✓ | △ |
| **Zero Padding** | ✓ | × | ✓ | N/A | N/A | ✓ | ✓ |
| **Length Prefix** | ✓ | × | ✓ | N/A | N/A | × | × |
| **Index Hints** | ✓ | × | ✓ | ✓ | × | N/A | N/A |
| **Abacus Emb.** | ✓ | × | △ | ✓ | × | N/A | N/A |

3. **Selection of Formatting/Indexing Augmentations**:

- *Reverse Digits* was retained as it is broadly compatible with N-Digit Chunking and directly addresses a known challenge for sequential models in arithmetic tasks, by processing numbers from least-significant to most-significant digit (Singh et al., 2024).

- *Zero Padding* vs. *Length Prefix*: Both techniques aim to provide information about number length or facilitate alignment across numbers of varying lengths. Zero Padding achieves this by pre-pending actual '0' digits to a fixed length before chunking, leveraging existing digit tokens. Length Prefix would involve new special tokens or a fixed-format prefix. As indicated in Table 3 by N/A (for their direct pairing, implying one subsumes or makes the other redundant) and × (for Length Prefix with Index Hints/Abacus, where explicit length tokens conflict with more granular positional encodings), these offer overlapping functionalities. We selected Zero Padding for its conceptual simplicity and direct integration with digit-based vocabularies.

- *Index Hints* vs. *Abacus Embeddings*: Both techniques aim to explicitly encode the positional significance of digits or chunks. Index Hints achieve this by associating specialized tokens or modifying embeddings based on a digit's (or chunk's) position. Abacus Embeddings are a specific type of learned positional embedding summed across digits. Table 3 marks their direct pairing as N/A as they serve the same fundamental purpose. We opted for the more general concept of Index Hints as it can be implemented flexibly and is readily combined with N-Digit Chunking.

This systematic filtering process, guided by logical compatibility and the goal of exploring diverse yet non-redundant strategies, led to our final set of tokenizer configurations for evaluation:

- Three N-Digit Chunking strategies (1-digit, 2-digit, 3-digit).

- Each of these chunking strategies is combined with four formatting/indexing options:

    1. None (i.e., only N-Digit Chunking is applied).
    2. Reverse Digits (applied to the number string before N-Digit Chunking).
    3. Zero Padding (applied to the number string before N-Digit Chunking).
    4. Index Hints (applied in conjunction with N-Digit Chunking).

    This yields 3 (chunking types) × 4 (formatting options) = 12 configurations.

- The standalone *xVal* configuration.

In total, this gives $12 + 1 = 13$ distinct tokenizer designs, forming a structured space for investigating the impact of tokenization on arithmetic reasoning, as detailed in the main paper (Section 3.1, Figure 1).

## E. Hyperparameter Optimization Strategies in YOTO

The selection of an optimal tokenizer for a Large Language Model (LLM) can be framed as a complex hyperparameter optimization (HPO) problem. The design space of tokenizers is vast, often combinatorial, and evaluating each candidate

typically requires expensive model training or significant fine-tuning (Feurer and Hutter, 2019). This section outlines the general landscape of HPO techniques, analyzes their applicability to the specific challenge of LLM tokenizer optimization, and details the reasoning that led to the HPO strategies integrated into our You Only Train Once (YOTO) framework.

### E.1. Overview of General Hyperparameter Optimization Techniques

HPO methods aim to find a set of hyperparameters that optimize a learning algorithm's performance. These techniques can be broadly categorized:

- **Model-Free Methods**: These approaches do not build an explicit model of the relationship between hyperparameters and performance.

  - *Grid Search*: Exhaustively evaluates all hyperparameter combinations on a predefined grid. While simple, it suffers from the curse of dimensionality.
  - *Random Search*: Samples hyperparameter configurations randomly from their respective distributions. Often more efficient than grid search, particularly when some hyperparameters are more influential than others (Bergstra and Bengio, 2012).

- **Model-Based Sequential Optimization (MBSO)**: These methods iteratively build a surrogate model (e.g., Gaussian Processes in Bayesian Optimization) to approximate the objective function and use an acquisition function to select the next hyperparameters to evaluate.

  - *Bayesian Optimization* (Snoek et al., 2012): Particularly effective for expensive black-box functions. Adaptations exist for combinatorial spaces (Baptista and Poloczek, 2018; Oh et al., 2019), relevant for discrete choices like tokenizers.
  - *Evolutionary Algorithms*: Employ principles of biological evolution, such as mutation, crossover, and selection, to iteratively refine a population of hyperparameter configurations.

- **Early Termination and Multi-Fidelity Optimization**: These techniques aim to reduce computational cost by quickly discarding unpromising configurations or by using cheaper, lower-fidelity approximations of the true evaluation.

  - *Successive Halving (SHA)* (Jamieson and Talwalkar, 2016): Allocates an initial budget to all configurations, evaluates them, and promotes only the top fraction (e.g., half) to the next round with an increased budget.
  - *HyperBand* (Li et al., 2018): Extends SHA by adaptively managing the number of configurations and the budget allocated at each stage, aiming for a good trade-off between exploration and exploitation. BOHB (Falkner et al., 2018) combines HyperBand with Bayesian Optimization.

- **Gradient-Based Methods for Differentiable Hyperparameters**: If hyperparameters are continuous and the objective function is differentiable with respect to them, gradient-based optimization can be used. For discrete hyperparameters, differentiable relaxations (e.g., Gumbel-Softmax (Jang et al., 2017; Maddison et al., 2017)) or implicit differentiation (Lorraine et al., 2020) can sometimes be applied.

- **Transfer Learning and Meta-Learning for HPO**: These approaches leverage knowledge from previous HPO tasks or related datasets/models to warm-start or guide the current optimization process (Feurer and Hutter, 2019). Predicting optimal hyperparameters for large models from smaller ones (e.g., $\mu$Transfer (Yang and Mahoney, 2022; Mahoney et al., 2024)) falls into this category.

### E.2. Challenges and Promising Directions for LLM Tokenizer HPO

Optimizing tokenizer choices for LLMs presents unique challenges:

1. **Discrete and Combinatorial Search Space**: Tokenizer design involves discrete choices (e.g., chunk size, vocabulary selection algorithm, formatting rules), leading to a vast and often non-ordered combinatorial space.

2. **Expensive Evaluations**: The primary bottleneck is the computational cost of training or extensively fine-tuning an LLM for each candidate tokenizer to assess its downstream performance.

3. **Interdependent Effects**: Tokenizer choices have a fundamental impact on the model's input data distribution and, consequently, on learned representations and task performance (Kaddour et al., 2023).

Given these challenges, several general HPO techniques face limitations:

- *Naive Grid/Random Search*: Becomes computationally intractable due to the high cost per evaluation and the size of the tokenizer search space.

- *Standard Gradient-Based Methods*: Not directly applicable for discrete, non-differentiable tokenizer choices, unless suitable differentiable relaxations of the tokenizer design space itself can be formulated, which is non-trivial.

- *Transfer Learning for Discrete Structures*: While promising for continuous hyperparameters (Yang and Mahoney, 2022), transferring discrete structural choices like tokenizers reliably across model scales or tasks is known to be difficult (Nguyen et al., 2023). Proxy models based on smaller model evaluations may not perfectly predict the ranking of tokenizers for larger models.

This analysis points towards HPO strategies that prioritize drastically reducing the evaluation cost per candidate or intelligently navigating the search space with fewer evaluations. Key promising directions include:

1. **Massive Reduction in Per-Candidate Evaluation Cost**:

   - *Shared Model Training / Resource Sharing*: If a single model can be trained in a way that allows evaluation or co-adaptation of multiple tokenizers simultaneously, the cost can be amortized. This involves designing a model architecture and training paradigm that can accommodate a diverse set of tokenization schemes. This is the central idea behind YOTO's unified vocabulary and shared LLM.

2. **Efficient Search and Early Pruning of Unpromising Candidates**:

   - *Early Termination Strategies*: Techniques like Successive Halving (SHA) are highly relevant. By allocating resources incrementally and discarding underperforming tokenizers early, the total computational budget can be significantly reduced.
   - *Model-Based Optimization for Parameterized Search*: If the choice among candidate tokenizers can be controlled by a learnable distribution (parameterized by $\phi$, as in YOTO), then techniques can be used to optimize these parameters. This transforms the discrete search problem into a potentially continuous optimization problem for the distribution's parameters.

### E.3. HPO in the YOTO Framework

The design of YOTO's optimization algorithm directly incorporates the insights discussed above, aiming to create an efficient and effective method for tokenizer selection:

1. **Unified Vocabulary and Shared LLM (??, Line 2)**: This is the cornerstone of YOTO's efficiency. Instead of training $C$ separate models for $C$ tokenizer candidates, YOTO trains a single LLM with a merged vocabulary encompassing tokens from all candidates. This drastically reduces the core model training cost, sharing the bulk of parameter updates across all considered tokenizers.

2. **Parameterized Exploration and Soft Successive Halving (??, Lines 3, 7, 13-21)**: YOTO manages a learnable categorical distribution, $p(S|\phi)$, over the candidate tokenizers $S \in \mathcal{T}$.

   - The parameters $\phi$ (logits for each tokenizer) are optimized during training, allowing the framework to learn which tokenizers contribute to better performance. The Gumbel-Softmax trick (Jang et al., 2017; Maddison et al., 2017) is employed to allow gradient flow back to $\phi$ despite the discrete sampling of a tokenizer $S_x$ for each training instance.
   - This learnable distribution is coupled with a "soft" variant of Successive Halving. Instead of rigidly discarding tokenizers, SHA in YOTO influences the logits $\phi$: tokenizers deemed unpromising by SHA (i.e., not in the top fraction after a rung evaluation) have their logits set to $-\infty$, effectively removing them from the sampling pool ($p(S|\phi) \approx 0$). This dynamically allocates training resources (data samples processed per tokenizer) towards more promising candidates.

3. **Alternative for $\phi$ Optimization (Mentioned in Main Paper)**: While the main paper focuses on Gumbel-Softmax for optimizing $\phi$, it also notes that for smaller, manageable design spaces, iterative methods like Bayesian Optimization, informed by the performance metrics ($m_S$) gathered during the SHA-like process, could offer a gradient-free alternative to update $\phi$ or directly select promising tokenizers.

By combining shared training with a resource-aware, adaptive exploration strategy based on soft SHA and parameterized tokenizer sampling, YOTO aims to discover high-performing tokenizers in sublinear time with respect to the number of candidates. This approach circumvents the prohibitive costs of naive evaluation while allowing the model and the tokenizer selection mechanism to co-adapt dynamically. The exclusion of direct proxy model transfer for tokenizer choice was due to its noted unreliability for discrete structural changes (Nguyen et al., 2023), favoring instead a joint optimization within a single, adaptable system.

## F. Synthetic Arithmetic Datasets and YOTO Training Setup

This section provides further details on the synthetic datasets generated for our arithmetic experiments and the hyperparameter configurations used for training the YOTO framework.

### F.1. Synthetic Dataset Generation for Arithmetic Tasks

For our experiments focusing on integer addition and scientific notation multiplication, we generated synthetic datasets to ensure controlled operand ranges and result distributions, facilitating a clear evaluation of tokenizer performance. The problem format is consistently operand_1 operator operand_2 = result (e.g., $A + B = C$ or $A * B = C$). Each dataset comprises 100,000 training samples and 10,000 test samples, as detailed in the main paper (Section 3.1).

The generation process for these datasets draws inspiration from the principles outlined in the Number Cookbook benchmark (Yang et al., 2024b), which aims for a comprehensive assessment of numerical understanding and processing abilities (NUPA) in LLMs. The Number Cookbook proposes a wide array of numerical tasks (41 combinations across 4 representations and 17 tasks) derived from educational curricula. Our work, while focused on demonstrating the YOTO framework's efficiency for tokenizer selection, uses a simplified subset of these tasks—specifically integer addition and scientific notation multiplication—as sensitive testbeds where tokenization choices have a pronounced impact.

While Number Cookbook strives for exhaustive coverage of numerical reasoning facets (e.g., various digit lengths, specific number properties, multi-step reasoning in some of its complex tasks), our synthetic datasets for YOTO experiments are tailored to clearly isolate the effects of tokenization on fundamental arithmetic operations. For instance:

- **Operand Characteristics**: We control operand lengths and ensure a diverse distribution of values, following similar principles to Number Cookbook to avoid trivial cases or biases. For problems with two operands, their lengths are varied to test alignment capabilities, as described in the main paper.

- **Task Simplification**: Our 'A op B = C' format directly tests the model's capacity to process the input numbers according to the specified operator and produce the correct output. This contrasts with some Number Cookbook tasks that might be embedded in more complex natural language instructions or require intermediate reasoning steps. This simplification is intentional: YOTO's primary contribution is an efficient tokenizer selection methodology, not a new state-of-the-art in broad numerical reasoning. The chosen tasks provide a clear signal for tokenization performance without confounding factors from more complex reasoning.

- **Focus on Core Arithmetic**: Integer addition is a foundational arithmetic skill, while scientific notation multiplication tests handling of more structured numerical formats (mantissa, exponent) and can be sensitive to how numbers are segmented and represented. These serve as representative tasks where different tokenization strategies (e.g., digit-wise, xVal, formatting) can lead to significantly different model performances.

The streamlined dataset design for YOTO is thus sufficient and appropriate for its objective: to demonstrate that YOTO can efficiently identify high-performing tokenizers for tasks where tokenization is critical, without the need for the full breadth of a comprehensive NUPA benchmark during the HPO process itself.

## F.2. YOTO Experimental Hyperparameter Setup

The YOTO framework (Algorithm 1 in the main paper) involves several key hyperparameters for both the shared LLM training and the Successive Halving Algorithm (SHA) based tokenizer selection process. The primary model architecture used for demonstrating YOTO is a Transformer-based model with 104M parameters, adhering to the principles of SuperTinyLMs (Hillier et al., 2024). All models are trained from scratch on NVIDIA L40s GPUs using bfloat16 precision.

**Shared LLM Training Hyperparameters:**

- **Optimizer**: AdamW (Loshchilov and Hutter, 2019) with $\beta_1 = 0.9$, $\beta_2 = 0.999$, and $\epsilon = 10^{-8}$.

- **Learning Rate for LLM parameters** ($\lambda_\theta$): $3 \times 10^{-4}$, with a cosine decay schedule and a linear warmup of 2,000 steps.

- **Learning Rate for Tokenizer Logits** ($\lambda_\phi$): $1 \times 10^{-4}$, also with a cosine decay schedule and linear warmup.

- **Batch Size**: A global batch size of 1024 sequences.

- **Total Training Steps** ($T_{\text{max}}$ **for YOTO**): The YOTO framework is trained for a total of 100,000 steps. This number is chosen to be comparable to the training duration of a single baseline model, allowing for substantial computational savings as detailed in Table 2 of the main paper.

- **Vocabulary**: A unified vocabulary is constructed from the union of all tokens across the 13 candidate tokenizers. Duplicate tokens are merged.

- **Gumbel Temperature** ($\tau$): For the Gumbel-Softmax reparameterization used to sample tokenizers, the temperature $\tau$ is annealed from an initial value of 2.0 down to 0.5 over the first 50% of the total training steps, and then kept constant.

**Soft Successive Halving (SHA) Hyperparameters for Tokenizer Pruning:** The SHA mechanism within YOTO dynamically manages the set of active tokenizers ($T_{\text{active}}$) by periodically evaluating and pruning less promising candidates based on their validation performance ($m_S$, typically validation loss contribution or exact match accuracy on a held-out set).

- **Candidate Tokenizers** ($T$): 13 distinct tokenizer configurations as described in Section 3.1 of the main paper.

- **Initial Logits** ($\phi_0$): Uniformly initialized for all 13 candidates, ensuring equal sampling probability at the start.

- **Reduction Factor** ($\eta$): $\eta = 3$. In each SHA rung, roughly $1/\eta$ of the currently active tokenizers are promoted.

- **Number of Rungs**: With $C = 13$ candidates and $\eta = 3$, there are $\lceil \log_\eta C \rceil = \lceil \log_3 13 \rceil = 3$ rungs (or pruning stages).

- **Initial Budget per Candidate** ($r_0$ **for first rung evaluation**): The first SHA evaluation and potential pruning occur after each active tokenizer has processed an average of approximately 15,000 training samples (tracked by $b_S$). This budget $r_{rung}$ increases by a factor of $\eta$ for subsequent rungs. ($r_0 \approx 15k$, $r_1 \approx 45k$, $r_2 \approx 135k$ cumulative samples per surviving tokenizer before next evaluation, though total training is capped by $T_{max}$).

- **Performance Metric for Pruning** ($m_S$): Primarily validation exact match accuracy on a dedicated validation set, updated periodically (e.g., every 5,000 training steps).

- **Pruning Mechanism**: When a SHA rung evaluation is triggered (i.e., surviving tokenizers in $T_{\text{active}}$ have their $b_S \geq r_{\text{rung}}$), the corresponding logits $\alpha_S$ for pruned tokenizers are set to $-\infty$, effectively removing them from being sampled by $p(S|\phi)$.

This setup ensures that YOTO efficiently explores the tokenizer design space, focusing computational resources on promising candidates while leveraging a shared model to co-adapt model parameters and tokenizer selection.

