# OpenReview forum: "You Only Train Once: Efficient Tokenizer Selection for Arithmetic in Language Models"
_ICML.cc/2025/Workshop/TokShop — TokShop_

### Official Review · Reviewer_szgd · 2025-06-07
**Review of YOTO: A Novel and Computationally Efficient Framework for Co-Adapting LLMs and Tokenizers**

**Rating:** 8
**Confidence:** 4

**Review:**

**Summary of the Paper**

This paper introduces You Only Train Once (YOTO), a unified training framework designed to address the computationally prohibitive challenge of selecting optimal tokenizers for Large Language Models (LLMs). Traditional methods require training a separate model for each candidate tokenizer, which is infeasible for large-scale models. YOTO tackles this by jointly optimizing the language model and a parameterized distribution over candidate tokenizers during pretraining.
The core of YOTO involves training a single LLM with a merged vocabulary that encompasses all tokens from all candidate tokenizers. During training, a learnable categorical distribution, parameterized by φ, governs which tokenizer is applied to each training instance, allowing for adaptive sampling of tokenizations. To enable gradient-based optimization for this discrete sampling, YOTO utilizes the Gumbel-Softmax reparameterization. Furthermore, YOTO integrates a soft variant of the Successive Halving Algorithm (SHA) to enhance efficiency, dynamically allocating training resources and prematurely terminating less promising tokenizer candidates by setting their sampling probability to effectively zero.
The framework is empirically validated on arithmetic reasoning tasks, specifically integer addition and scientific notation multiplication, which serve as a challenging testbed where tokenization significantly impacts performance. The paper defines a structured design space of 13 number tokenization strategies, combining options for representation, chunking (1-digit, 2-digit, 3-digit, xVal), and ordering/formatting/indexing (None, Reverse, Padding, Index Hints). Performance is evaluated using exact match accuracy, mean absolute error, and mean relative error, comparing YOTO against a baseline of independently training models for each tokenizer from scratch.

**Strengths:**
* Addresses a Critical Computational Bottleneck: YOTO directly confronts the significant challenge and computational expense of tokenizer selection, which traditionally requires full model retraining for each candidate. This is a major practical hurdle in LLM development.
* Exceptional Computational Efficiency: The framework demonstrates dramatic computational savings, reducing Floating Point Operations (FLOPs) by over 80% for a single model run compared to baseline training. This saving is further amplified when considering the need to evaluate multiple tokenizer candidates, as YOTO only requires a single training run to find the optimal tokenizer, scaling sub-linearly with the number of candidates.
* Principled Joint Optimization: YOTO offers a novel and principled approach to co-adapt language models and tokenizers dynamically within a unified training paradigm. This allows for a more integrated and effective design process.
* Strong Empirical Validation: The method effectively discovers high-performing number tokenizers for arithmetic tasks. The results show high correlation between YOTO's rankings and ground truth rankings, particularly for top-performing tokenizers, ensuring that the optimal solutions are identified efficiently.
* Broad Applicability: While focused on numerical tokenization, the paper emphasizes that the YOTO method is broadly applicable and can be extended to explore larger and more diverse tokenizer search spaces, including morphological, multilingual, or byte-level schemes, as well as being applied to larger models and varied modalities beyond arithmetic.
* Clear Design Space and Methodology: The paper provides a well-defined and reasoned structured design space for number tokenization, along with detailed explanations of the choices made and the compatibility of various techniques. The algorithm (Algorithm 1) and its components are clearly described.

**Weaknesses:**
* Focused Empirical Scope: The empirical validation is specifically centered on arithmetic tasks and uses Transformer-based models up to 104M parameters. While acknowledged as deliberate for rigorous testing within a well-defined space, extending YOTO to general language complexities and much larger, frontier LLMs (e.g., 13B, 70B+) is noted as future work and remains an open challenge.
* Design Space Generalization: The structured design space for arithmetic tokenization, while comprehensive, might not directly translate to general language tokenization. Developing similar structured design spaces for broader vocabulary elements is a "substantial research challenge".
•
Optimization Technique Exploration: While effective, YOTO's use of soft Successive Halving and Gumbel-Softmax could potentially be further improved by exploring a broader range of hyperparameter optimization algorithms or more advanced early-termination methods, particularly for larger candidate sets.

---

### Official Review · Reviewer_Pvrx · 2025-06-08
**YOTO - Joint Optimization of Tokenizer and Model**

**Rating:** 8
**Confidence:** 4

**Review:**

The paper introduces YOTO (You Only Train Once), a novel framework for efficiently selecting optimal tokenizers for LLMs. The paper addresses an important computational bottleneck and shows promising results on arithmetic tasks. The core idea captured in the paper is to jointly optimize language model parameters and tokenizer selection in a single training run. It proposes to merge multiple candidate tokenizers' vocabularies which can have limitations in practice.

**Strengths:**
- The paper is well-written, content is easy to follow with visualizations that clearly communicate the technical details, methods, and results.
- Subject tackled in the paper, tokenizer selection is often overlooked but can have substantial impact on model performance, especially on structured tasks like Arithmetic. The task used for evaluation is appropriate.
- The idea introduced, joint optimization of tokenizer and model parameters, is novel and theoretically sound.
- Motivates the need from tokenizer optimization by sharing relevant historical studies.
- Empirical evidence presented demonstrates that the YOTO algorithm effectively identifies right tokenizers and hence has potential for practical applications (needs further studies on other tasks)
- The authors rightly captured the limitations of the proposed approach

**Weaknesses:**
- The evaluation is limited to one structured task (arithmetic) with relatively simple tokenization strategies. It is unclear how the approach generalizes to other complex natural language tasks with more diverse tokenizer space, large model space and multi-modal applications. In addition, the model used in relatively small (104M) compared to current SOTA models, more exploration is required to understand if the efficiency gains hold at the sota scale.
- The unified vocabulary framework proposed requires merging all candidate tokenizers' vocabularies. In practice, this can create very large vocabularies and the computation/memory implications of this approach should be addressed
- Need more details on when the approach could perform poorly. The lower correlation for multiplication tasks hints at potential limitations that require further analysis

Overall, YOTO is a sound technical contribution that opens up interesting research direction, but more extensive validation (especially across diverse tasks and at scale) are required to establish its practical utility for practical LLM development.

---

### Decision · Program_Chairs · 2025-06-10

Accept